# Spot Invalid Point Repair Algorithm of Detector Array Measurement System Based on Image Correlation Coefficient

Yilun Cheng [1,2,3], Gangyu Wang [1,2,3], Fengfu Tan [1,3], Feng He [1,3], Laian Qin [1,3], Zhigang Huang [1,3] and Zaihong Hou [1,3,*]

1　Key Laboratory of Atmospheric Optics, Anhui Institute of Optics and Fine Mechanics, Hefei Institutes of Physical Science, Chinese Academy of Sciences, Hefei 230031, China; cyl1008@mail.ustc.edu.cn (Y.C.); wgy0606@mail.ustc.edu.cn (G.W.); tff@aiofm.ac.cn (F.T.); fhe@aiofm.ac.cn (F.H.); laqin@aiomfm.ac.cn (L.Q.); huangzhigang@aiofm.ac.cn (Z.H.)
2　Science Island Branch of Graduate School, University of Science and Technology of China, Hefei 230026, China
3　Advanced Laser Technology Laboratory of Anhui Province, Hefei 230037, China
*　Correspondence: zhhou@aiofm.ac.cn; Tel.: +86-173-5658-4808

**Abstract:** The detector array method has been widely used in the field of high-energy laser far-field spot parameter measurement due to its ability to directly measure the far-field spot of high-energy lasers, wide dynamic range of the detectors, high system sampling frequency, good real-time performance, and suitability for various testing environment requirements. However, during the measurement process, the irradiation of strong lasers or damage to other hardware systems can result in invalid points in the acquired spot images, thereby reducing the measurement accuracy of the system. In order to achieve accurate measurement of laser far-field spot parameters, this paper establishes an experimental model based on the analysis of the sampling spacing of the detector array target and proposes a laser far-field spot invalid point repair algorithm based on image correlation coefficient. Experimental results demonstrate that the algorithm proposed in this paper effectively reduces the impact of invalid points in the measurement system on the measurement accuracy, and achieves accurate measurement of high-energy laser measurement systems.

**Keywords:** detector array method; spot parameter; spot invalid point; image correlation coefficient

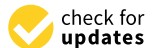



## 1. Introduction

The detector array method is used to arrange the detectors sensitive to specific wavelengths according to certain rules to form a detector array target. The detectors distributed on the target surface generate electrical signals through the photoelectric effect after being irradiated with the incident laser, and then perform digital conversion processing on the electrical signals to obtain the measurement data of the temporal and spatial distribution of laser intensity [1–3]. The laser intensity distribution parameters mainly include total laser power, spot center of mass position coordinates, spot diameter, beam quality factor $\beta$, and so on. By obtaining these parameters, on the one hand, the experimental data can be used to carry out the study of the laser atmospheric transmission effect; on the other hand, analysis of the laser far-field spot parameters can be used to evaluate the performance parameters of the laser launching system. The detector array method is more and more widely used in laser parameter measurement technology because of its direct measurement of the spot, wide dynamic range of the detector, high system sampling frequency, and good real-time performance, and can be applied to various test environment requirements [4–6]. The hardware architecture is shown in Figure 1.

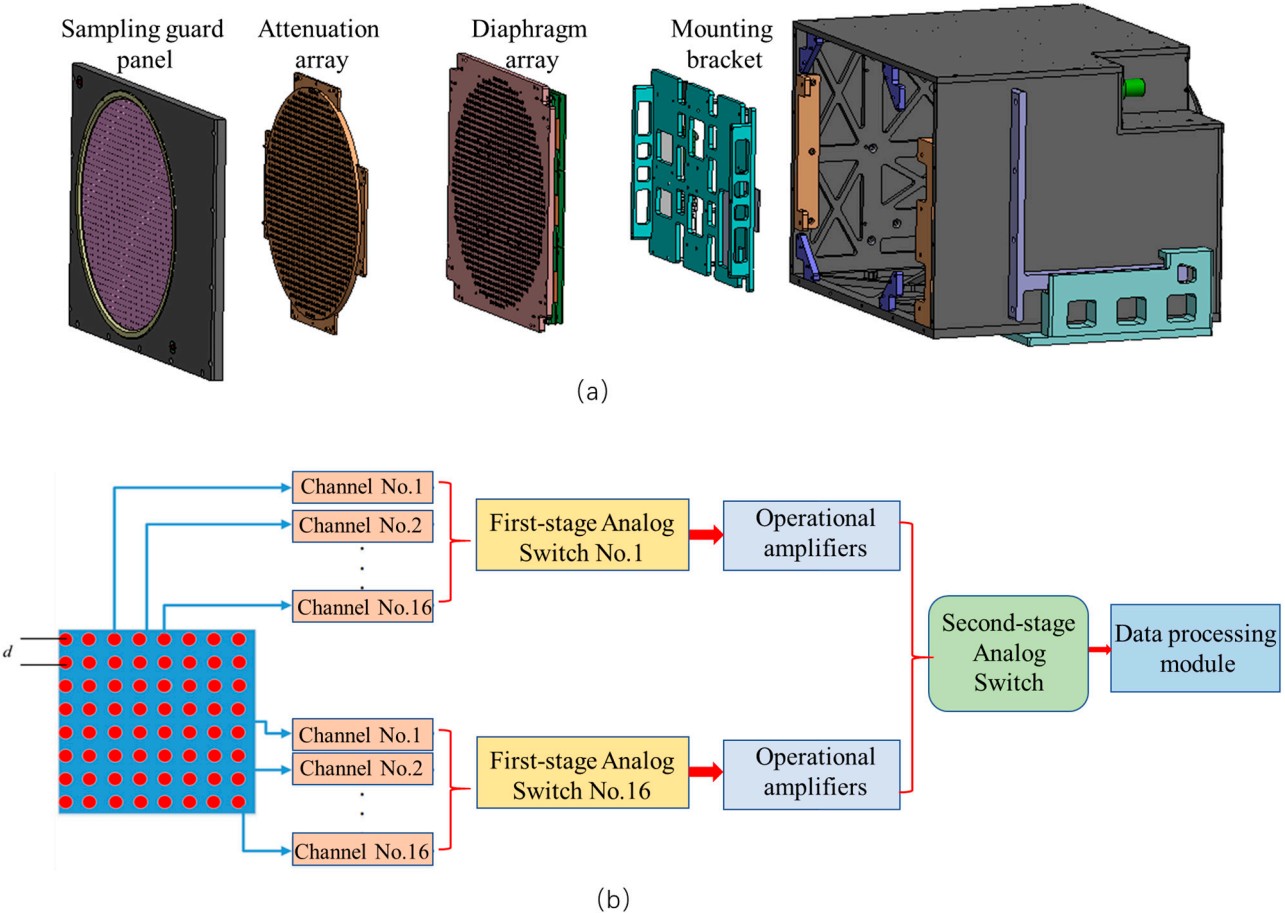

**Figure 1.** Structure diagram of detector array method: (**a**) Mechanical structure diagram. (**b**) Electrical structure diagram.

It can be seen from Figure 1 that a large number of photodetectors are installed on the surface of the detector array target, and each detector is independent of the other on the target surface; that is, when one detector fails, other detectors will not be affected. Several detectors are connected to a first-stage analog switch (generally 16 to 1). When the first-stage analog switch fails, the 16 detectors connected to it cannot work normally, and 16 failure points are generated at 1 time. The signal passing through the first-order analog switch enters the second-order analog switch through the operational amplifier. A second-order analog switch controls 256 detectors. When the second-order analog switch fails, 256 failure points will be generated on the target surface, which is fatal to the measurement accuracy of the system. Therefore, the reliability of the system is often improved by backup on the second-order analog switch. On the other hand, when the high-energy laser irradiates the surface of the measurement system to generate a failure point, the failure point model generated at this time is also a continuous regional failure point in most cases. Therefore, only the continuous failure point problem of adjacent regions is discussed in this paper. When the measurement system contains regional failure points, the generation of failure points obtained with the measurement system will lead to a decrease in the accuracy of the measurement data. Therefore, in this case, it is necessary to repair and reconstruct the data at the failure point position in the system.

At present, image inpainting technology is divided into two categories. One is digital image inpainting technology for repairing image defects in small-scale space [7–11]. This method uses the edge information of the area to be repaired in the damaged image. At the same time, it also uses the method from rough calculation to accurate calculation to estimate the direction of the image's equal illuminance line and uses the propagation

mechanism to propagate the information of the intact area in the image to the area to be repaired, and finally realizes the image inpainting of the damaged area. The other is the image completion technology for filling in the lost information of large blocks in the image. At present, this kind of technology also includes the following two methods. One is the repair technology based on image decomposition. The main idea is to decompose the image into structural parts and texture parts. The structural part is repaired using the inpainting algorithm, and the texture part is filled using the texture synthesis method. The other is an image completion algorithm for repairing the image information of large area block damage points in the image [12–16]. At present, this algorithm also includes two main methods. The first is the repair technology based on image decomposition. The main idea of this method is to decompose the image to be repaired to obtain the structural part of the image and the part of the image texture. The inpainting algorithm is used to repair the structural part, and the texture synthesis method is used to fill the texture part. Another method is to use texture synthesis technology based on image blocks to fill in the missing information [17–21]. This algorithm first selects a pixel point from the boundary of the damaged image repair area that needs to be repaired, and then takes this pixel point as the center to select the appropriate size of the texture block according to the texture features of the image. Finally, the closest texture matching block around the damaged area must be found to replace the texture block in this area, and completion of the repair of the damaged area can take place.

## 2. Materials and Methods

### 2.1. Calculation Method of Spot Parameters in Array Detection Method

According to the requirement of repair and reconstruction of failure points in high-energy laser far-field parameter measurement systems, this paper first introduces the relevant theoretical basis of laser parameter measurements based on the detector array method and the specific calculation method of spot parameters. Then, a digital simulation model is established to analyze the influence of failure points in the measurement system on the measurement accuracy of the system. Finally, according to the correlation of the image in the image sequence collected with the measurement system and the correlation of the energy coefficient, a failure point repair and reconstruction algorithm based on the combination of image structure similarity and spot energy coefficient is proposed.

In the detector array method, the total laser energy can be obtained with Formula (1).

$$P(f) = \sum_{i=1}^{m} \sum_{j=1}^{n} A_{ij} I_{ij}(f) \tag{1}$$

where $m$ represents the number of rows of the detector array, $n$ represents the number of columns of the detector array, and $A_{ij}$ represents the area represented by the detection unit with rows and columns $(i, j)$ (in cm$^2$). In the subsequent research work of this paper, the total power of the spot is characterized by the total ADU (Analog-to-Unit) of the system.

The meaning of the diffraction limit factor $\beta$ is the ratio of the far-field divergence angle of the measured actual laser beam to the far-field divergence angle of the ideal beam. In the detector array method measurement system, its expression can be expressed with Formula (2).

$$\beta = \theta\_real / \theta\_ideal \tag{2}$$

where $\theta\_real$ is expressed as the far-field divergence angle of the laser beam actually measured, and $\theta\_ideal$ is the divergence angle of the laser beam under ideal conditions. For the centroid position parameters of the spot, assuming that the acquisition resolution of the measurement system is $X \times Y$, and the pixel value of each point on the image is represented by $G(x,y)$, the position coordinates of the spot centroid can be obtained with Formula (3).

$$\begin{cases} x_0 = \dfrac{\sum\limits_{x=1}^{X}\sum\limits_{y=1}^{Y} G(x,y)\times x}{\sum\limits_{x=1}^{X}\sum\limits_{y=1}^{Y} G(x,y)} \\[2em] y_0 = \dfrac{\sum\limits_{x=1}^{X}\sum\limits_{y=1}^{Y} G(x,y)\times y}{\sum\limits_{x=1}^{X}\sum\limits_{y=1}^{Y} G(x,y)} \end{cases} \tag{3}$$

### 2.2. Failure Point Repair Algorithm

The flow chart of the region failure point repair and reconstruction algorithm based on image similarity is shown in Figure 2. Firstly, the position of the failure point is obtained and marked according to the position location algorithm of the failure point in the repair process, and then the template image of the failure point position is obtained by extending several pixels outward with the center position of the failure point. During the expansion process, all positions in the failure point area are included in the template image. Then, starting from the upper left corner of the first frame image of the spot image sequence, the matching template is obtained with the same size as the failure point template. In the internal area of the template, the pixel value is set to zero according to the position of the failure point, and the matching template is slid with a step size of one pixel until all areas of the original image are traversed. After obtaining the matching templates at different positions, the correlation coefficient between the template and the failure point template is calculated according to the image similarity calculation formula [22]. The optimal matching block is found by calculating the correlation coefficient. The optimal matching block is filled according to the location area of the failure point found in the above steps to complete the repair of the failure point in the image of the failure block. This process starts from the first frame of the sequence image to the end of the last frame.

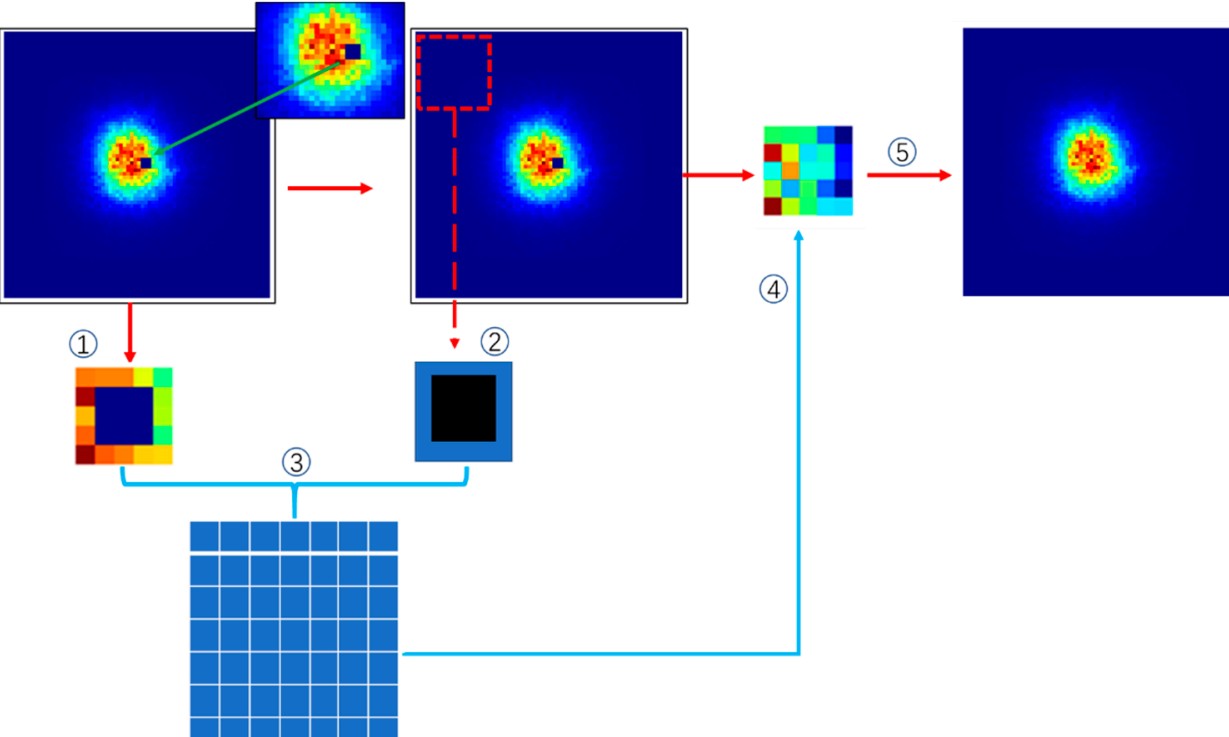

**Figure 2.** Spot image restoration algorithm based on structural similarity.

For the repair and reconstruction of the failure point of the detector array target measurement system, the first step is to extract the position information of the failure point according to the image information. When the failure point is a static failure point, the response value at its location is a fixed value, which is shown as a dark point or a saturation point on the image. When the failure point is a dynamic failure point, the response value is normal within a certain range, but when it exceeds a certain threshold, the response is not normal. In this paper, the dynamic threshold algorithm is used to detect the location of the failure point [23]. First, the integral spot image is filled with two rows and two columns to the outside, and the image is traversed with a $3 \times 3$ template. At each position, the 8 pixel values around the center of the template are sorted, and the pixel values of the two points in the middle are taken as the weighted mean, which is recorded as Value0. Then, the 8 adjacent pixel values are reordered in the order from large to small. In the sorting process, the maximum value, the minimum value, and the middle two values are removed, which are recorded as $value_1$, $value_2$, $value_3$, and $value_4$, respectively. Then, the dynamic threshold $k$ at this position can be calculated with Formula (4).

$$k = \frac{(value_1 + value_2) - (value_3 + value_4)}{2} \tag{4}$$

After obtaining the dynamic threshold k, according to Formula (5), it can be judged whether the point is a failure point. If the pixel value $g(x,y)$ of this position is within its range, it can be considered as a normal response point; otherwise, it can be regarded as a failure point.

$$value_0 - k \leq g(x,y) \leq value_0 + k \tag{5}$$

After obtaining the position information of the failure point with the above method, the failure point position template shown in step 1 of Figure 2 can be obtained by spreading a pixel outward from the center point, which is recorded as G1. The image is traversed with a template of the same size to obtain a matching template. Each matching template is assigned a value according to the pixel value at the position of its failure point, and the matching template in step 2 shown in Figure 2 can be obtained, denoted as G2. After obtaining the failure point position template G1 and the matching template, it is necessary to calculate the correlation between the two. The methods for calculating the image correlation mainly include an evaluation method based on the histogram correlation coefficient, evaluation method based on mean square error, evaluation method based on peak signal-to-noise ratio and evaluation method based on image structure similarity. However, in the correlation calculation of the laser spot image, the similarity of the two images cannot be simply considered, and the difference in the total pixel value of the two images and the difference in the structure should be considered. The calculation method based on image structure similarity has this characteristic. Therefore, when calculating the correlation between the failure point location template and the matching template, the image structure similarity is used as a measure to evaluate its similarity. The evaluation method of image structural similarity fully considers the three-feature information of brightness, contrast, and structural degree between images. The structural information of images is defined as a structural attribute independent of image brightness and contrast, and the distortion model is used as a combination of image brightness, image contrast, and image structural degree. The calculation method is shown in Formulas (6)–(9).

$$l(x,y) = \frac{2\mu_x\mu_y + C_1}{\mu^2{}_x + \mu^2{}_y + C_1} \tag{6}$$

$$c(x,y) = \frac{2\sigma_x\sigma_y + C_2}{\sigma^2{}_x + \sigma^2{}_y + C_2} \tag{7}$$

$$\delta(x,y) = \frac{\sigma_{xy} + C_3}{\sigma_x \sigma_y + C_3} \tag{8}$$

$$SSIM(x,y) = \frac{(2\mu_x \mu_y + C_1)(2\sigma_x \sigma_y + C_2)}{(\mu^2{}_x + \mu^2{}_y + C_1)(\sigma^2{}_x + \sigma^2{}_y + C_2)} \tag{9}$$

where $x$ represents the template image, $y$ is the contrast image, $\mu_x$ and $\mu_y$ are the mean values of the two images, $\sigma_x$ and $\sigma_y$ are the standard deviations of the images, $\sigma_{xy}$ is the covariance of the two images, and $C_3 = C_2/2$ is in the simplified mode. By calculating the failure point template and the matching template, the correlation coefficient matrix in step3 of Figure 2 can be obtained. Since the value range of $x$ is between 0 and 1, the stronger the correlation between the two images, the closer the calculation result is to 1. Therefore, the restoration and reconstruction of the pixel value of the failure point position can be realized by retrieving the coefficient matrix. It is worth noting that when the failure point position in the image is not completely filled after one repair, the above steps need to be repeated until the pixel values at all failure point positions are completely filled.

## 3. Results

### 3.1. System Sampling Resolution Analysis

The detector array method samples the incident laser beam through the detection unit, transforms the far-field spot image into a discrete dot matrix signal on the two-dimensional plane, and finally processes the discrete dot matrix signal through the data processing unit to obtain the spatial and temporal distribution intensity parameters of the incident beam. Due to the limitation of space physics and the development cost of the measurement system, the detector sampling units on the target surface cannot be closely arranged together. The spacing distance between two adjacent detectors in Figure 1 is expressed as the sampling resolution of the system. When the incident laser is irradiated on the target surface, the photoelectric response occurs at the position where the target surface contains the detector. The optical signal is converted into a digital signal and then stored and calculated in the form of an image matrix. Due to the limitation of research and development costs and physical size, there is a certain interval between detectors. Therefore, when analyzing the influence of failure points on measurement accuracy, it is necessary to establish a reasonable system sampling interval.

In this section, while aiming to investigate the influence of sampling resolution on the measurement accuracy of the system in the detector array target measurement system, a laser transmission model under three different turbulence intensities is constructed with the laser emission system output wavelength of 1064 nm; the laser transmission distance of L = 2 km; the laser emission aperture of $D$ = 400 mm; the laser emission system transmission beam quality factor $\beta_0$ = 3 in vacuum; and the atmospheric coherence length of 2 cm, 4 cm, 6 cm, and 8 cm at the wavelength of 1064 nm. The far-field spot images under different turbulence intensities are sampled with different sampling intervals to obtain the sampled spot, and the sampled spot image is interpolated to make the resolution of the spot image the same as the resolution of the original spot. Finally, the spot parameters described in Section 2.1 of this paper are used for the calculations, respectively, and then the influence of the sampling interval of the measurement accuracy in the measurement system is analyzed. A total of 100 frames of original spot images are generated and transmitted through different turbulence intensity environments, and integral spot images under different turbulence intensities are obtained, as shown in Figure 3. The pixel resolution of the spot image is 512 pixels × 512 pixels. The pixel size of the single pixel is 0.92 mm.

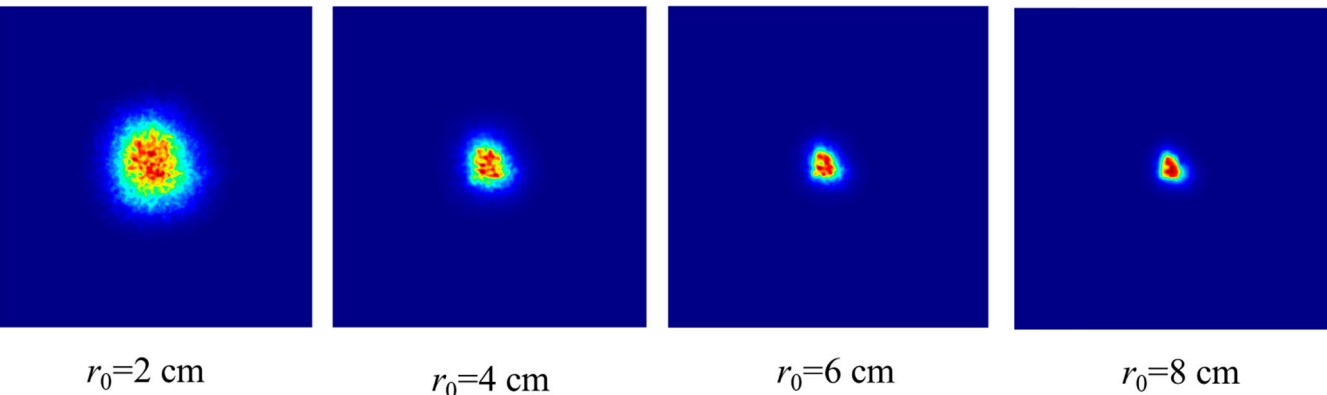

$r_0$=2 cm $\qquad$ $r_0$=4 cm $\qquad$ $r_0$=6 cm $\qquad$ $r_0$=8 cm

**Figure 3.** Integral spot of different turbulence intensities.

The original spot in the above image is downsampled according to different sampling intervals, and then the sampled discrete spot is interpolated to obtain the restored spot image. According to the calculation method described in Section 2.1 of this paper, the original spot and the restored spot are processed and compared to obtain the relationship between the system measurement accuracy and the detector array sampling resolution. Simulation experiments are carried out according to different sizes of spots and sampling resolutions. Therefore, the whole experiment is divided into spot downsampling processing, spot image restoration, and spot parameter calculation.

When sampling the spot images of different sizes, the influence of different sampling points on the calculation error of laser parameters must be considered. Under the premise of ensuring that the target surface can receive all the spots, the 100 frames of spot sequence images are downsampled at the sampling intervals of 3.68 mm, 4.66 mm, 5.52 mm, 6.44 mm, 7.36 mm, 8.28 mm, and 9.20 mm, respectively, to obtain the discrete sampling spots. After obtaining the spots with different sampling resolutions, the bilinear interpolation algorithm is used to interpolate the sampling spots. The pixel size of the interpolated image is consistent with that of the original spot.

After obtaining the restored spot images under different sampling resolutions, the original spot and the restored 100-frame spot images are calculated to obtain the relationship between the spot parameter error data under different turbulence intensities and the system sampling spacing. The instantaneous spot parameters of 100 frames calculated above are counted, and the statistical results are expressed as a standard deviation. The error relationship curve between the spot parameters and the sampling interval is shown in Figure 4.

It can be seen from Figure 4 that, when the turbulence intensity on the laser transmission path is constant, the spot parameter error calculated with the measurement system increases with the increase of the sampling interval. When the sampling interval of the system is less than 4 mm, the calculated spot ADU peak error, spot centroid position coordinate error, and spot total ADU error decrease with the decrease of turbulence intensity. When the sampling interval of the system is greater than 4 mm, the calculated parameter errors increase with the decrease in turbulence intensity. This is due to the fact that when the turbulence intensity is large, the spot diameter from the far field to the target spot is continuously expanded and the degree of fragmentation inside the spot is more intense. When the sampling spacing is small, the system can detect smaller speckles, thereby reducing the system measurement error. However, the calculated error is close to the real spot. From the above analysis, it can be seen that it is reasonable to select the sampling interval of 5.52 mm.

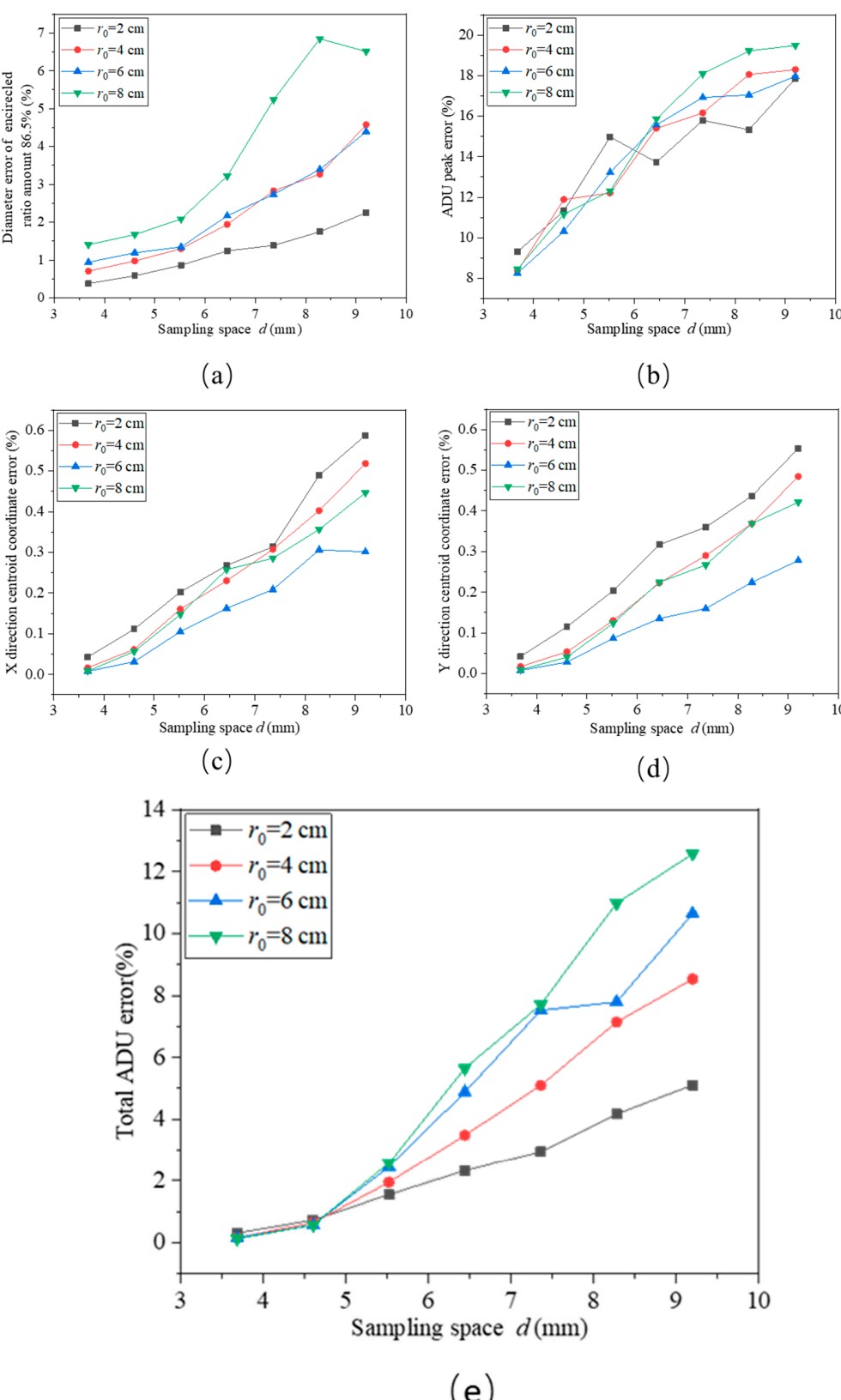

**Figure 4.** Error curve between spot parameters and sampling space: (**a**) Error curve between spot diameter and sampling space. (**b**) Error curve between spot peak ADU and sampling space. (**c**) Error curve between X-direction centroid coordinates and sampling space. (**d**) Error curve between Y-direction centroid coordinates and sampling space. (**e**) Error curve between spot total ADU and sampling space.

### 3.2. Failure Point Repair

Based on the above analysis, we sample the spot in Figure 3 with a sampling space of 5.5 mm, and add a failure point to the sampling spot. According to the algorithm proposed in this paper, the image containing the failed spot is repaired and reconstructed. In addition, the control group processes them with the mean filtering algorithm [24], respectively, and the obtained spot image is shown in Figure 5.

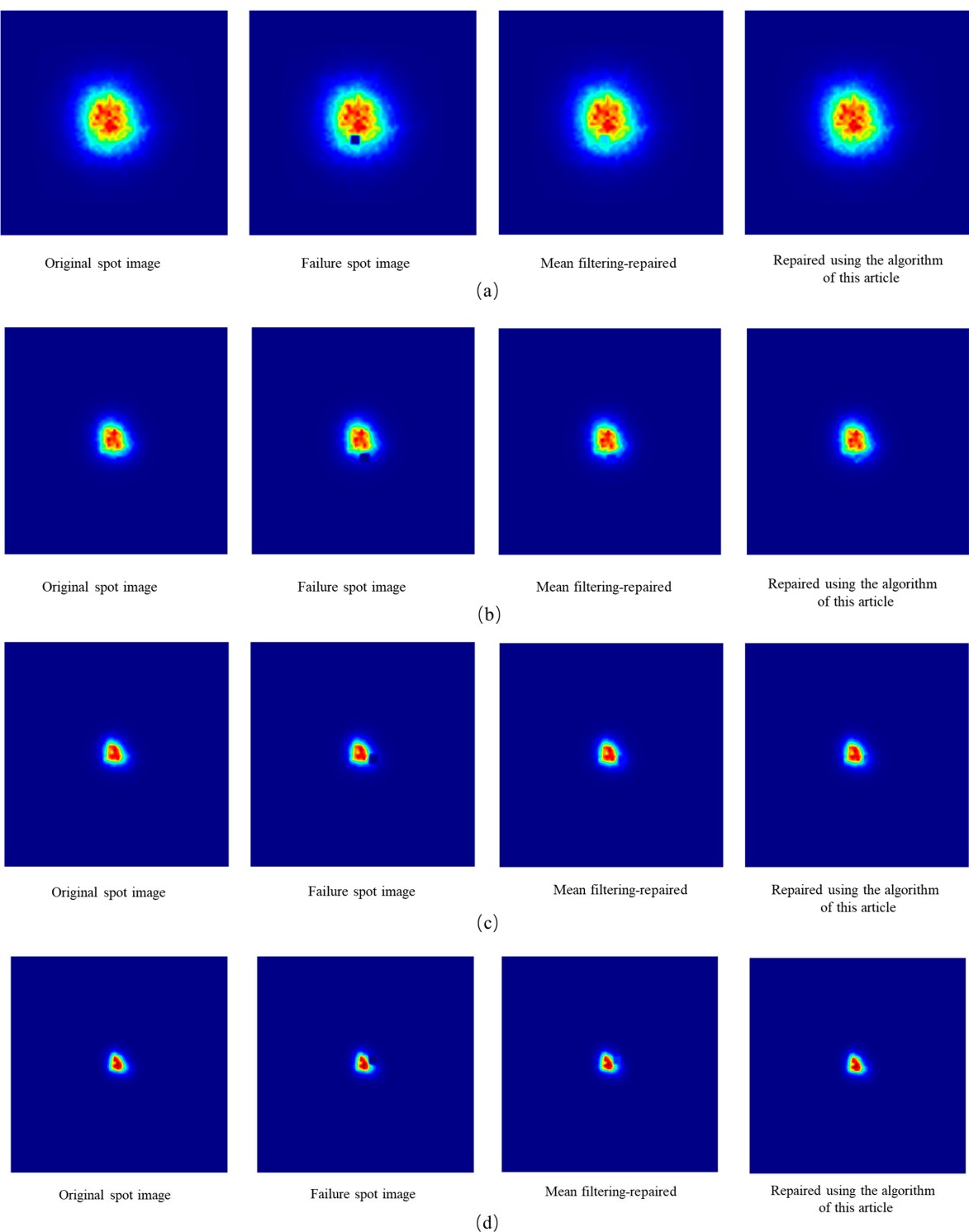

**Figure 5.** Spot images processed with different methods. (**a**) $r_0 = 2$ cm; (**b**) $r_0 = 2$ cm; (**c**) $r_0 = 4$ cm; (**d**) $r_0 = 8$ cm.

It can be seen from the above figures that, when there is a single regional failure spot module in the measurement system, the algorithm proposed in this paper can effectively repair and reconstruct the failure spot data, which is intuitively superior to the data processing algorithm of error spot filtering. In addition, the spot image processing under different turbulence intensities fully proves the robustness of the algorithm.

When several regional failure points appear on the target surface, the area occupied by the failure points may be close to or even larger than the area occupied by the spot because the detectors are arranged at certain intervals on the target surface. Therefore, it is not very meaningful to analyze the data of weak turbulence in this case. In order to verify the influence of the existence of multiple failure points on the calculation accuracy of the measurement system, the turbulence spot image when the atmospheric coherence length $r_0$ equals 2 cm is taken as the object of analysis and multiple failure point modules are set up in the sampling template, which are processed using the algorithms proposed in this paper and the algorithm based on mean filtering. The spot image is obtained as shown in Figure 6.

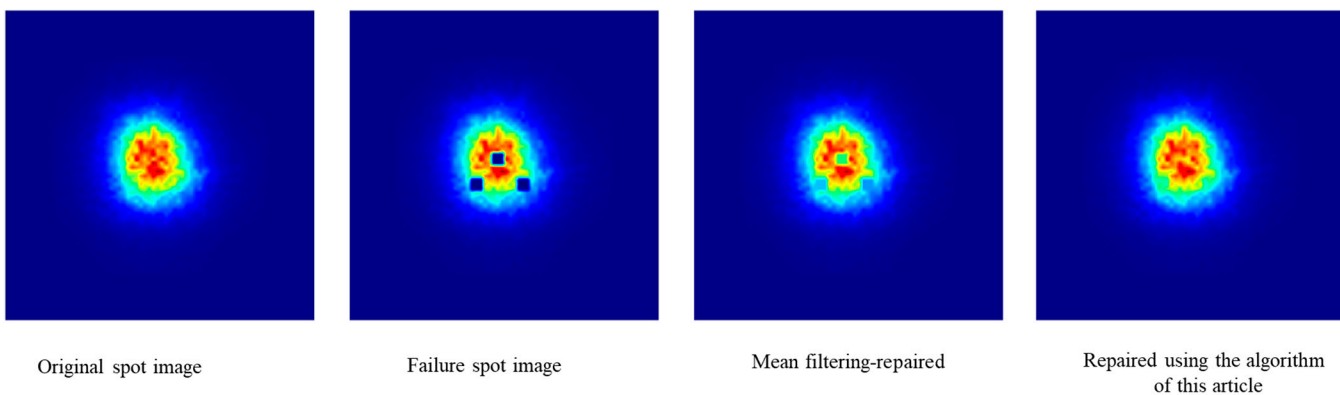

Original spot image    Failure spot image    Mean filtering-repaired    Repaired using the algorithm of this article

**Figure 6.** Repaired images under multiple failure points.

As can be seen from Figure 6, the algorithm proposed in this paper still performs well when there are several failure point modules in the light spot image.

## 4. Discussion

The laser parameter calculation method of the detector array measurement system described in Section 2.1 was used to calculate the spot shown in Figure 6, and the obtained spot parameters are shown in Table 1.

**Table 1.** The calculated spot parameters.

|  | Total ADU | Centroid Coordinates | 86.5% Encircled Energy Diameter |
|---|---|---|---|
| $r_0$ = 2 cm | | | |
| Original image | 1,339,177 | (226.60, 230.10) | 220 |
| Contains failure points image | 1,314,216 | (226.72, 229.32) | 222 |
| Processed using mean filtering algorithm | 1,329,823 | (226.64, 229.82) | 220 |
| Processed using the algorithm in this article | 1,338,153 | (226.60, 230.07) | 220 |
| $r_0$ = 4 cm | | | |
| Original image | 1,336,776 | (230.62, 232.58) | 120 |
| Contains failure points image | 1,305,693 | (230.42, 231.71) | 122 |
| Processed using mean filtering algorithm | 1,327,363 | (230.56, 232.36) | 120 |
| Processed using the algorithm in this article | 1,338,174 | (230.63, 232.67) | 120 |

**Table 1.** *Cont.*

|  | Total ADU | Centroid Coordinates | 86.5% Encircled Energy Diameter |
|---|---|---|---|
| $r_0 = 6$ cm |  |  |  |
| Original image | 1,337,314 | (232.13, 233.69) | 88 |
| Contains failure points image | 1,268,275 | (230.80, 233.08) | 90 |
| Processed using mean filtering algorithm | 1,309,551 | (231.68, 233.47) | 90 |
| Processed using the algorithm in this article | 1,318,116 | (231.81, 233.56) | 88 |
| $r_0 = 8$ cm |  |  |  |
| Original image | 1,339,797 | (232.97, 234.37) | 72 |
| Contains failure points image | 1,265,863 | (232.02, 234.74) | 74 |
| Processed using mean filtering algorithm | 1,330,345 | (233.10, 234.28) | 72 |
| Processed using the algorithm in this article | 1,338,480 | (232.87, 234.39) | 72 |

It can be seen from Table 1 that the existence of the failure point has a great influence on the total ADU value of the laser spot obtained with the measurement system, and has little influence on the other parameters. When the atmospheric coherence length varies from 2 cm to 4 cm, the total ADU error is 1.86%, 2.33%, 5.16%, and 5.52%; after repairing the failure point with the mean filtering algorithm, the ADU error is reduced to 0.6%, 0.7%, 2.1%, and 0.7%. When the failure point is processed with the algorithm in this paper, the ADU error is reduced to 0.07%, 0.1%, 1.4%, and 0.09%.

## 5. Conclusions

To sum up, while aiming to investigate the failure point problem in the detector array laser parameter measurement system, we propose a repair algorithm based on the similarity of image structure between frames. By processing the spot images under different atmospheric transmission conditions, the accuracy of the failure point correction averaging filtering algorithm is significantly improved, which proves that the algorithm has strong reliability and robustness. The proposed algorithm can be well applied to the failure point repair of the detector array measurement system.

**Author Contributions:** Conceptualization, Y.C. and G.W.; methodology, F.T. and Z.H. (Zhigang Huang); software, F.H. and Y.C.; writing—original draft preparation, Y.C. and G.W.; writing—review and editing, F.T., Z.H. (Zaihong Hou), and L.Q. All authors have read and agreed to the published version of the manuscript.

**Funding:** This research received no external funding.

**Institutional Review Board Statement:** Not applicable.

**Informed Consent Statement:** Not applicable.

**Data Availability Statement:** Data are available upon request.

**Conflicts of Interest:** The authors declare no conflict of interest.

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
