# Peer review of "Spot Invalid Point Repair Algorithm of Detector Array Measurement System Based on Image Correlation Coefficient"

_photonics, doi:10.3390/photonics10101105_

Round 1

Reviewer 1 Report

1.      How did the authors explain that the existence of the failure point has a great influence on the total ADU value of the laser spot other than other parameters?

2.      The authors mentioned that it is worth noting that when the failure point position in the image is not completely filled after one repair, the above steps need to be repeated until the pixel values at all failure point positions are completely filled. Is it possible the failure point position in the image could not be completely filled all the time?

NA

Author Response

Response to Reviewer 1 Comments

Point 1: How did the authors explain that the existence of the failure point has a great influence on the total ADU value of the laser spot other than other parameters?

Response 1: Thanks for your careful checks. First, the failure point affects the measurement parameters of the system only when the incident laser irradiates in the region of the failure point of the measurement system, and the data loss at the failure point is a case where the pixel value at the spot image is 0. Therefore, the effect of the failure point inside the spot on the position of the spot center of mass is not particularly noticeable; second, the spot diameter at the spot annular periphery power ratio of 86.5% is an integral quantity ( The ratio of the sum of the power inside different diameters to the total power to the target spot), which also reduces the effect of the failure point on it to a certain extent; finally, for the peak power, because the proposed algorithm fixes the object of the integrated spot over a period of time, the failure point will have a great effect on the power of the spot only when the failure point falls in the peak position of its power, otherwise, the failure point's impact on the peak value can be nearly not be counted. The above is also supported by Figure 4 in Section 3 of this paper.

Point 2: The authors mentioned that it is worth noting that when the failure point position in the image is not completely filled after one repair, the above steps need to be repeated until the pixel values at all failure point positions are completely filled. Is it possible the failure point position in the image could not be completely filled all the time?

Response 2 : Thanks for your careful checks. The failure point repair algorithm based on image structure similarity proposed in this paper is to utilize a large amount of data in the sequence of light spot images for repairing, and each frame is calculated to select the optimal matching block and then integrate it so as to realize the repair of the failure point data of the integral light spot. Since the laser transmission in the atmosphere is disturbed by turbulence and the aiming deviation or tracking deviation of the transmitting system itself will lead to the change of the spot position, while the position of the failure point is fixed, the incomplete repair of the failure point region basically does not occur.

Reviewer 2 Report

I got acquainted with the work "Spot invalid point repair algorithm of detector array measure-ment system based on image correlation coefficient" by a group of scientists led by Yilun Cheng. While reading the work, I had a lot of questions and comments.

1. In line 157, 160 and 185, most likely a typo, since figure 7 is not in the article.

2. If I understood correctly, then the proposed algorithm obtained was tested on real data. If so, then I have a lot of questions:

a. The wavelength of the radiation is indicated, but the radiation power is not indicated.

b. The article is about "turbulence". I would like to know the parameters that characterize it.

c. I understand correctly that the averaging was over 100 frames? What is the registration frequency. What method was used to average the data.

d. What is the fluctuation of the intensity of the laser spot and its position?

e. And how do all these factors affect the operation of the algorithm?

3. From the description of the algorithm, I did not quite understand how the process of restoring intensity in the damaged area of the image takes place? Based on bilinear interpolation algorithm? Then why use correlation in general here? To search for this image corruption? But what happens if the "pixel" of the image is overexposed? I ask the authors to clarify this point.

4. What will happen to the algorithm as a whole if there is a high level of background noise?

5. The cases shown in Figure 6 are very simple. How will the algorithm behave if the damaged area is in the center of the spot?

6. What, according to the author, does this algorithm have disadvantages?

7. What happens if there is more than one damaged area in the image?

Author Response

Response to Reviewer 2 Comments

Point 1: 1. In line 157, 160 and 185, most likely a typo, since figure 7 is not in the article.

Response 1: Thanks for your careful checks. We have changed figure 7 to figure 2.

Point 2:If I understood correctly, then the proposed algorithm obtained was tested on real data. If so, then I have a lot of questions:

  1. The wavelength of the radiation is indicated, but the radiation power is not indicated.

Response a : Thanks for your careful checks. In this paper we characterize laser power in terms of ADU, which is characterized as a relative quantity. For example, when the ADU value is equal to 1, we can convert it to 1 w, 10 w, or even 100 w. Therefore, we use ADU to characterize the laser power.

  1. The article is about "turbulence". I would like to know the parameters that characterize it.

Response b : Thanks for your careful checks. Turbulence is when the flow velocity of a viscous fluid reaches a certain value and the whole fluid begins to move irregularly with the material. The actual atmosphere is not a homogeneous optical medium and is always in constant flow. The atmospheric temperature varies randomly over small areas and short periods of time due to the effects of wind shear formed by the drag of the airflow by the earth's surface, uneven heating of the earth's surface by the sun, and convection caused by thermal irradiation of the earth's surface. Random variations in atmospheric temperature produce variations in atmospheric density, which in turn leads to variations in atmospheric refractive index. The cumulative effect of this correlation results in significant differences in atmospheric refractive index. The turbulence discussed in atmospheric optics refers to the random variation in atmospheric refractive index. When a laser is transmitted through a turbulent atmosphere, the turbulence affects it mainly in the following ways: light intensity scintillation, beam drift, beam extension, reach angle undulation, focal plane spot dispersion, and so on. The characteristic parameters that characterize the intensity of atmospheric turbulence are mainly the atmospheric coherence length r0 and the refractive index structure constant , and these two parameters can be converted to each other.

  1. I understand correctly that the averaging was over 100 frames? What is the registration frequency. What method was used to average the data.

Response c : Thanks for your careful checks. First, in this paper we constructed 100 frames of spot image data under different turbulence intensities using a numerical simulation program that The spot images are sampled by the detector array model containing the failure point, so the average number of frames in the failure point repair simulation experiments in this paper is 100 frames. Secondly, the original spot sequence is computed at a frequency of 25 Hz, and finally, the instantaneous spot images are computed to find the optimal matching region during the repair process, and integrated in the time domain to compute the average value to be used as the repair data of the integrated spot. The description of this part may not be very clear, and it has been modified in the latest manuscript.

  1. What is the fluctuation of the intensity of the laser spot and its position?

Response d : Thanks for your careful checks. When the laser is transmitted through the atmosphere the change of turbulence intensity will inevitably cause the undulation of light intensity, position change and morphological change. In this paper, the spot intensity fluctuations and center of mass position changes under different turbulence intensities are described as shown in the figure below.

Figures. 1 Center-of-mass position curve (a) r0=2 cm; (b) r0=4 cm; (c) r0=6 cm; (d) r0=8 cm;

Figures 2. Curves of ADU undulation (a) Total ADU; (b) ADU peak;

  1. And how do all these factors affect the operation of the algorithm?

Response e : Thanks for your careful checks. When the laser is transmitted to the far field through the atmosphere, the turbulence perturbation will lead to the distortion of the far-field spot image, which mainly includes the rise and fall of the intensity, the change of the position, and the change of the morphology. For the two factors of intensity ups and downs and the shift of spot center of mass position, the intensity ups and downs and the position of center of mass do not have much effect on the integral spot image because it is still approximately Gaussian distributed. However, the change of the spot image size caused by the change of turbulence may affect the recovery accuracy of the algorithm to some extent. Therefore, in this paper, four spot images with different turbulence intensities are selected for calculation, and the results prove that the restoration accuracy of the algorithm proposed in this paper is still much better than that of the restoration algorithm based on median filtering.

Point 3. From the description of the algorithm, I did not quite understand how the process of restoring intensity in the damaged area of the image takes place? Based on bilinear interpolation algorithm? Then why use correlation in general here? To search for this image corruption? But what happens if the "pixel" of the image is overexposed? I ask the authors to clarify this point.

Response 3: Thanks for your careful checks. The failure point repair algorithm proposed in this paper is shown in Fig 3. Firstly, the failure point in the integrated spot is detected by the dynamic thresholding method and its location is labeled. Then the failure point template is obtained by spreading a number of pixels around the location of the failure point and is denoted as G1, followed by constructing a matching template of the same size as the failure point template starting from the first frame of the image and is denoted as G2. There are many matching templates in a frame, so we choose the algorithm of the image structural similarity to compute the G1 and G2, so as to select the optimal matching block, which is computed frame-by-frame until the last frame of the image sequence. The process is computed frame by frame until the end of the last frame of the sequence. Finally, the matching blocks calculated in each frame are accumulated and averaged in the time domain, and the resulting data are used as the restoration data at the location of the failure point. The mean filtering algorithm exists in the sense that it appears as a pair of controls, by comparing the algorithm proposed in this paper with the algorithm of mean filtering.

Figures 3. Spot image restoration algorithm based on structural similarity.

Point 4. What will happen to the algorithm as a whole if there is a high level of background noise?

Response 4: The first question: in the laser parameter intensity distribution measurement system, the crucial point is to ensure that the measurement system to obtain the spot image will not appear saturated, but the spot image saturation will lead to a significant reduction in the credibility of the measurement data, then at this point in time the test data are required to be discarded. Therefore, the data processed in the detector array measurement system is not saturated. The Second question: First, the laser power measured in the laser parameter intensity distribution measurement system is high, so the signal-to-noise ratio of the image is high. Secondly, the noise floor will be reduced by hardware circuit optimization during the design of the measurement system. Finally, the image data are processed for noise reduction after acquisition.Therefore, the background noise of the image will not have a large impact on the algorithm proposed in this paper.

Point 5. The cases shown in Figure 6 are very simple. How will the algorithm behave if the damaged area is in the center of the spot?

Response 5: Thanks for your careful checks. A simulation test was added to the results of the article to simulate the failure point in the center region of the spot. In addition, Figure 6 in the original text was incorrectly numbered and has been corrected

Point 6. What, according to the author, does this algorithm have disadvantages?

Response 6: Thanks for your careful checks Any algorithm will have certain limitations, the main idea of the algorithm proposed in this paper is to use the atmospheric turbulence laser far-field spot produced by jitter, morphological changes and other factors affect the repair. Therefore, the first point, when the failure point is located in the center of the spot and the turbulence is weak, the algorithm is almost impossible to complete the repair of the spot image; the second point is that, when the size of the spot to the target is small and the target surface appears a number of failure point area, in this case the proposed algorithm will not be very good repair effect.

Point 7. What happens if there is more than one damaged area in the image?

Response 7: Thanks for your careful checks.A simulation for different failure point locations have been added to the results in this paper.

Reviewer 3 Report

The authors aimed to utilize an experimental model based on the analysis of the sampling spacing of the detector array target and proposed a laser far-field spot invalid point repair algorithm based on image correlation coefficient. Their results showed that the failure points processed by the algorithm have substantially reduced to below 1% or so. These findings could ultimately be applied to the failure point repair of the detector array measurement systems. This issue is important in a wide range of laser far-field measurements. Overall, I find the manuscript well-written. The algorithm sounds reasonable, and the experimental results seem technically correct. Therefore, I would recommend publication in its present form despite one minor question:

What are the error bars for the data points in Figure 4? Are they taken from a single measurement or average from many shots?

Minor editing of English language required.

Author Response

Response to Reviewer 3 Comments

Point 1: What are the error bars for the data points in Figure 4? Are they taken from a single measurement or average from many shots?

Response 1: Thanks for your careful checks. Figure 4 illustrates the error of each parameter obtained from the calculation, and its specific value is the result after comparing with the parameters obtained from the original spot calculation. And the results in Fig. 4 are calculated from 100 frames of spot images at different turbulence intensities respectively, So they average from many shots not taken from a single measurement.
